# Sequence Denoising with Self-Augmentation for Knowledge Tracing

## Abstract

Knowledge tracing (KT) aims to predict students' future knowledge levels based on their historical interaction sequences. Most KT methods rely on interaction data between students and questions to assess knowledge states and these approaches typically assume that the interaction data is reliable. In fact, on the one hand, factors such as guessing or slipping could inevitably bring in noise in sequences. On the other hand, students' interaction sequences are often sparse, which could amplify the impact of noise, further affecting the accurate assessment of knowledge states. Although data augmentation which is always adopted in KT could alleviate data sparsity, it also brings noise again during the process. Therefore, denoising strategy is urgent and it should be employed not only on the original sequences but also on the augmented sequences. To achieve this goal, we adopt a plug and play denoising framework in our method. The denoising technique is adopted not only on the original and the augmented sequences separately during the data augmentation process, but also we explore the hard noise through the comparison between the two streams. During the denoising process, we employ a novel strategy for selecting data samples to balance the hard and soft noise leveraging Singular Value Decomposition (SVD). This approach optimizes the ratio of explicit to implicit denoising and combines them to improve feature representation. Extensive experiments on four real-world datasets demonstrate that our method not only enhances accuracy but also maintains model interpretability.

## 1 Introduction

With the rise of online education, Knowledge tracing (KT) task has drawn wide concern and has become a major challenge (Embretson & Reise, 2000). It aims to predict the probability of a learner's mastery on the knowledge points based on the sequence of correct and incorrect responses across multiple historical learning tasks (Yin et al., 2023; Liu et al., 2023a; Long et al., 2021), enabling dynamic tracing of the learner's knowledge state.

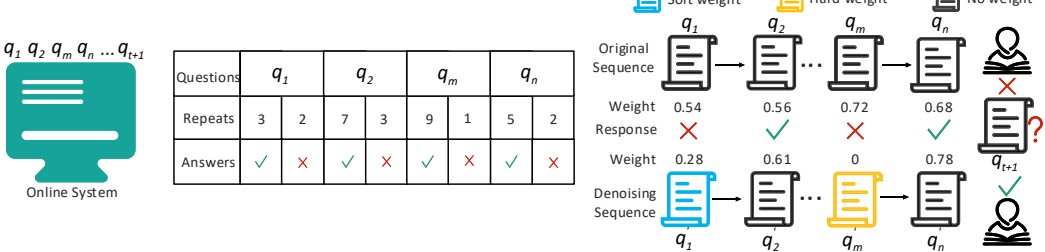

Figure 1: The difference in knowledge states between the original interaction sequence and the denoised sequence after multiple student responses, as well as the impact on future questions.

Although existing KT methods have achieved some success in forecasting students' future performance on questions (Wang et al., 2023; Corbett & Anderson, 1994), they are still influenced by noise. Taken Figure 1 as an example, in traditional KT methods, the influence weights learned from

the original interaction sequences on the final question may be incorrect due to some factors, *e.g.*, the mistake made by the carelessness of the student or the unreliable knowledge states. These outliers can be regarded as noise inevitable in interactions. Furthermore, sparse problem which always existed in interaction sequences could amplify the impact of noise and affect the representation of the students' final knowledge states. To address this issue, recent researches have utilized various deep models (Nakagawa et al., 2019; Wang et al., 2024; Liu et al., 2023b) to capture the sequentiality within sequences to against the risk of noise and employed contrastive learning for data augmentation to mitigate the problem of data sparsity. These approaches aim to uncover unique learning patterns or regularities among students, but they are always not involved in the noise generated in data augmentation, which is harmful to learn robust sequence representations, too.

In Figure 1, we assume that each question is answered some times. Question $q_1$ was answered correctly three times at first and incorrectly two times latter, indicating that $q_1$ has been gradually mastered. Based on this, the impact of the incorrect responses should be reduced on the target question. While for question $q_m$, it was answered correctly nine times and incorrectly once. This single incorrect response very likely might be a mistake, so we want to treat it as noise and ignore its impact on the target question. The comparison shows that the answers to the target question are completely different before and after denoising. Without denoising, the noise might lead to a decline in the student's knowledge states, resulting in incorrect answers. Besides noise, sparse is also an important problem in KT. Previous methods which leverage the data augmentation strategy may amplify the influence of noise due to the unreliability in original sequences, which is harmful to the performance of the model. Considering the two problems, we proposed the Sequence Denoising with Self-Augmentation for Knowledge Tracing method, which aims to address the noise in both the original sequences and augmented sequences from the explicit and implicit perspectives.

Specifically, after data augmentation to expand sparse interaction sequences, we employ a combined strategy of soft and hard denoising (Lin et al., 2023a; Yuan et al., 2021). Intuitively, the sample that is off center has the high probability is the outlier, that is to say, the noise usually has the high sharpness. To measure the degree of outlier, inspired by Singular Value Decomposition (SVD) (Zhai et al., 2024), we adopt the singular value to reflect the informative signals. Obviously, the larger the singular value, the more smooth. To make full use of the singular value, on the one hand, we maximize the largest singular value to reduce the sharpness, then the impact of noise could be weakend. On the other hand, to fully explore the noise, considering that the augmented data are randomly generated, the greater the feature difference, the more likely it is to be hard noise. Therefore, we not only leverage the SVD as the regularization to reduce the impact of noise, but also is helpful to the mining of noisy samples. Specifically, we apply SVD to explicitly explore the hard sequence data, while performing implicit denoising on the remaining data, and then merged the denoised sequences. This approach not only enhances the robustness of the model in noisy environments but also improves its interpretability.

The main contributions are summarized as follows. Firstly, we introduce a denoising module that combines explicit and implicit denoising methods and integrate them for the first time in the Knowledge Tracing (KT) task. This not only improves the prediction accuracy but also enhances the interpretability of the model. Secondly, our method is based on data augmentation, which has already been utilized in KT. We further utilize data augmentation to denoise both the original sequence and the augmented sequence and combine them to obtain a better data representation, which improves the reliability of the data while addressing the issue of data sparsity. Thirdly, we make use of Singular Value Decomposition (SVD) (Chen et al., 2019) not only for individual data sequences as a regularization term, but also to distinguish between explicit and implicit denoising samples, effectively extracting the underlying patterns and structures from the data and improving the data quality.

## 2 RELATED WORK

**Knowledge Tracing** KT methods were generally divided into two main categories: traditional machine learning methods and deep learning methods (Zhou et al., 2024; Abdelrahman et al., 2023). Among traditional approaches, Bayesian Knowledge Tracing (BKT) (Corbett & Anderson, 1994) stood out as a seminal method, leveraging the Hidden Markov Model (HMM) to sequentially model and interpret the student learning process. Other important methods included Performance Factors Analysis (PFA) (Pavlik et al., 2009) and Item Response Theory (IRT) (Reise, 2014), which focused

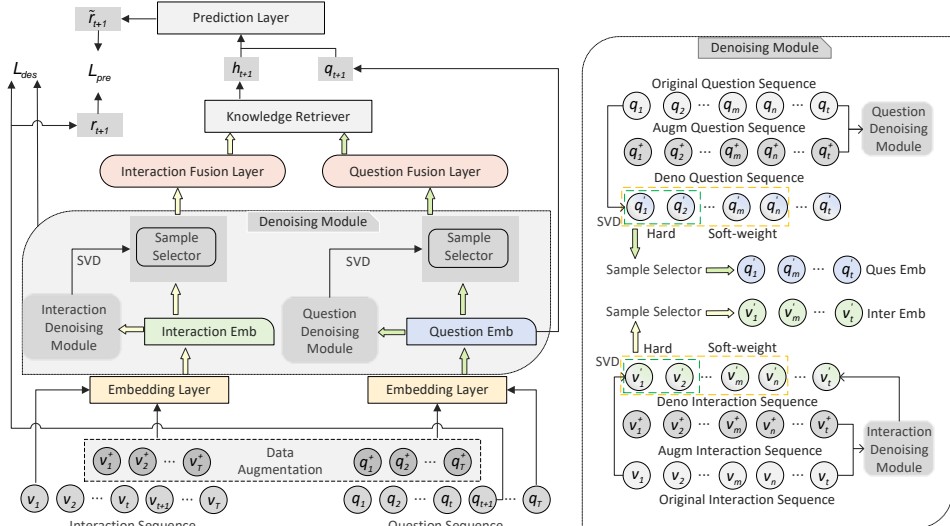

Figure 2: The overall framework of CL4KT-DA. The left side represents the overall model architecture, while the right side details the denoising module we proposed.

on different factors affecting performance. Recently, the advent of deep neural networks brought significant advancements with Deep Knowledge Tracing (DKT), which showed notable improvements in performance. Based on this, Self-Attentive Knowledge Tracing (SAKT) introduced attention mechanisms (Vaswani, 2017; Pandey & Karypis, 2019), allowing for the identification of correlations between different concepts and addressing data sparsity issues. Furthermore, contrastive learning for Knowledge Tracing (CL4KT) (Lee et al., 2022; Liu et al., 2020a) incorporated contrastive learning methods (Robinson et al., 2021) to enhance historical interaction sequences through effective data augmentation (Dang et al., 2023; Liu et al., 2020b). However, previous methods have used data augmentation to tackle data sparsity, they have not fully addressed the potential noise introduced by such augmentation. Addressing noise within sequences remains an often-overlooked issue, which can significantly affect model performance.

**Sequence Denoising** Recent KT studies have explored many methods to learn better feature representations. However, in practice, historical sequences usually contain some inherently noisy items (such as guessing or slipping) (Zhang et al., 2023), which are always ignored, resulting in inaccurate final predictions. Although few approaches utilize denoising to achieve better data representation in KT field, the denoising methods are widely applied in sequence-related tasks, which could inspire our research. Some studies addressed this challenge in a "soft" way (Zhang et al., 2022), trying to implicitly reduce the noise on the learned sequence representation, *i.e.*, assigning lower weights to those interactions that are less important relative to the final interactions, but in this way, noise still existed in sequence and may affect performance. Furthermore, some studies directly use explicit denoising (Tong et al., 2021; Han et al., 2023) to delete irrelevant items in the sequence. However, the historical sequence usually contains some interactions that are irrelevant to the next interaction, which may not be inherent noise, so eliminating them without carefully selected may lose useful information. Inspired by these observations, we want to combine explicit and implicit denoising, balancing the influence of primary and secondary features on the next interaction. This strategy not only ensures performance but also enhances the model's interpretability.

## 3 TWO-STREAM DENOISING MODEL

Figure 2 offers a detailed overview of the CL4KT-DA model, which integrates the CL4KT framework with our denoising module. Notably, we employed only the data augmentation from CL4KT.

## 3.1 PROBLEM DEFINITION

The student's historical learning interactions are defined as $V$, where $V = (v_1, v_2, ..., v_m, v_n, ..., v_t)$. Each interaction consists of a tuple: $v_t = (q_t, r_t)$. Where $q_t$ represents the $t_{th}$ question, and $r_t$ represents the response result, which is either 0 or 1, with 1 indicating a correct response and 0 indicating an incorrect response. Given the interaction sequence and the next question $q_{t+1}$, KT aims to determine the probability of correctly answering the next question:

$$\hat{r}_{t+1} = p(r_{t+1} = 1 | v_1, v_2, ..., v_t, q_{t+1}). \tag{1}$$

## 3.2 MODEL ARCHITECTURE

**Data-Augmentation** Due to the complexity and uniqueness of the KT task, directly applying existing data augmentation methods from CV (Chen et al., 2020; Hochreiter & Schmidhuber, 1997) and NLP (Gao et al., 2021) is challenging. Therefore, we follow the data augmentation approach in contrastive learning and use various data augmentation methods to generate relevant views of students' learning histories. These methods include: 1. Question Masking: Replacing some questions in the original history with a special mask without changing their responses. 2. Interaction Cropping: Randomly extracting a subsequence from the original history. 3. Question Replacement: Transforming the original question into a simpler or more difficult one based on its response. 4. Interaction Shuffling: Reordering interactions within a subsequence of the original history. Each of these data augmentation methods is applied with different probabilities. Ultimately, this results in augmented question sequences: $Q_1^+ = (q_1^+, q_2^+, ..., q_m^+, q_n^+, ..., q_t^+)$ and interaction sequences: $V_1^+ = (v_1^+, v_2^+, ..., v_m^+, v_n^+, ..., v_t^+)$.

**Embedding Layer** We initially map the original questions and interactions, as well as the augmented IDs, to dense embedding vectors $q_{u_i}, q_{u_i}^+, v_{u_i}$ and $v_{u_i}^+ \in \mathbb{R}^s$, where $s$ represents the dimension of the embedding vectors and $W_q$, $W_{q^+}$, $W_v$ and $W_{v^+}$ are the trainable matrices. Consequently, the embeddings of the questions and interactions are initialized as follows:

$$q_{u_i} = q_i W_q, \quad v_{u_i} = v_i W_v.$$
$$q_{u_i}^+ = q_i^+ W_{q^+}, \quad v_{u_i}^+ = v_i^+ W_{v^+}. \tag{2}$$

**Denoising Module** To obtain better sequence representations, we applied data augmentation to both question and interaction sequences. However, since the original sequences may contain inherent noise, the same denoising process was applied to both the augmented and original sequences. On one hand, the augmented and original sequences usually share similar data distributions and noise characteristics. On the other hand, the augmented sequences provide richer and more diverse data, helping prevent the model from overly focusing on specific noise patterns. This approach allows us to effectively remove noise while preserving data diversity, thereby improving the quality of sequence representations. In the denoising process, we adopted the denoising method $f_{den}$ proposed in (Zhang et al., 2022; Lin et al., 2023b). This method utilizes information from both augmented and original sequences to generate noiseless sub-sequences through a specific denoising mechanism. Specifically, $f_{den}$ filters noise by leveraging intra-sequence information:

$$q_d = f_{den}(q_{u_i} | q_i, \Theta_{d_q}), v_d = f_{den}(v_{u_i} | v_i, \Theta_{d_v}),$$
$$q_d^+ = f_{den}(q_{u_i}^+ | q_{i^+}, \Theta_{d_q}), v_d^+ = f_{den}(v_{u_i}^+ | v_{i^+}, \Theta_{d_v}). \tag{3}$$

where $\Theta_{d_q}$ and $\Theta_{d_v}$ represent the parameters of $f_{den}$. To identify the main learning patterns and rules in historical interactions and extract more accurate features, we combine explicit denoising with implicit denoising, unlike previous approaches that considered only one type of denoising. Using only explicit denoising can eliminate inherent noise but might mistakenly remove interactions with low similarity to the target interaction. On the contrary, using only implicit denoising merely reduces the weights of unrelated interactions, leading to incomplete noise removal. In our method, we make use of the fusion of the denoised augmented sequence obtained from Eq.(3) with the denoised sequence to obtain new problem and interaction sequences:

$$q_d^{'} = q_d + \lambda \cdot q_d^+, \quad v_d^{'} = v_d + \lambda \cdot v_d^+. \tag{4}$$

where $\lambda$ is a trade-off parameter to balance the contribution of the augmented sequence when generating the final sequence representation.

In order to ensure that the explicit and implicit denoising samples we selected have high representativeness and signal-to-noise ratio, we need to select a sample subset for weighted fusion. To ensure that the model is not adversely affected by implicit noise, inspired by (Chen et al., 2019), we incorporate an SVD-based loss function into the training process to softly reduce the noise. This loss is specifically designed to reduce the impact of noise, allowing the model to focus more on capturing general features and latent patterns while enhancing data quality.

$$\mathcal{L}_{des} = -\frac{\delta_1}{\sum_{j=1}^{D} \delta_j}. \tag{5}$$

where $\delta_j$ represents the maximum singular value, $D$ denotes the size of the singular value matrix.

Besides the implicit denoising, due to the singular value could reflect the informative signal, SVD is also leveraged to the explicit denoising process. We use it to select a sample subset and quantify the denoising effect, thereby identifying samples that retain the main information and remove noise. By comparing the original sequences and the augmented sequences, the higher the difference, the higher the probability of noise. Therefore, we first convert the original problem and interactive embeddings as well as the problem and interactive embeddings after noise reduction into a matrix and decompose them to obtain the reconstruction features:

$$qm_i = U_{q_i} \cdot \Sigma_{q_i} \cdot V_{q_i}^{\top}, \quad qm_d^{'} = U_{q_d^{'}} \cdot \Sigma_{q_d^{'}} \cdot V_{q_d^{'}}^{\top}. \tag{6}$$

$$vm_i = U_{v_i} \cdot \Sigma_{v_i} \cdot V_{v_i}^{\top}, \quad vm_d^{'} = U_{v_d^{'}} \cdot \Sigma_{v_d^{'}} \cdot V_{v_d^{'}}^{\top}. \tag{7}$$

Then, we calculate the singular value difference of the question and interaction matrices respectively. The larger the difference value, the less similar the features are, and classify this as noise data. We compare the original embeddings of questions and interactions with the denoised embeddings, looking for significant differences between matrices. We select samples with larger differences for explicit denoising, and the remaining samples with smaller differences for implicit denoising. This approach avoids excessive denoising and ensures data integrity. To choose an appropriate threshold, we integrate the differences of the question and interaction to measure the signal-to-noise distribution. The higher the noise, the higher the threshold $\rho$.

$$\Delta_{ques} = \alpha \cdot \frac{||\Sigma_{q_d}||_2 - \left|\left|\Sigma_{q_d^{'}}\right|\right|_2}{\max(||\Sigma_{q_d}||_2, \left|\left|\Sigma_{q_d^{'}}\right|\right|_2)} + (1-\alpha) \cdot (1 - \cos(\theta_{q_d, q_d^{'}})), \tag{8}$$

$$\Delta_{inter} = \beta \cdot \frac{||\Sigma_{v_d}||_2 - \left|\left|\Sigma_{v_d^{'}}\right|\right|_2}{\max(||\Sigma_{v_d}||_2, \left|\left|\Sigma_{v_d^{'}}\right|\right|_2)} + (1-\beta) \cdot (1 - \cos(\theta_{v_d, v_d^{'}})), \tag{9}$$

$$\Delta_{global} = \gamma \cdot \Delta_{ques} + (1-\gamma) \cdot \Delta_{inter}, \tag{10}$$

$$\rho = \mu(\Delta_{global}) + k \cdot \sigma(\Delta_{global}) \cdot H(\Delta_{global}). \tag{11}$$

$\alpha$ and $\beta$ are hyperparameters used to measure the proportion of the influence of singular values and eigenvectors. $q_d, q_d^{'}$ and $v_d, v_d^{'}$ are the angles between the corresponding eigenvector spaces. $\gamma$ represents the proportion of weights of problems and interactions. Samples are classified according to the value of. Samples with high values correspond to those significantly affected by noise and will undergo explicit denoising. $H$ represents the calculation of information entropy, and $k$ is a regulating coefficient used to measure the overall uncertainty of the data.

Specifically, we choose the top- $\lfloor \rho/4 \rfloor$ samples for explicit denoising and we use the mask to explicitly denoise the sampled data. In order to balance the main and secondary features, the remaining sequence is fused with the original sequence for implicit denoising, resulting in the final representation of the question and interaction:

$$\text{mask}_q[j] = \begin{cases} 1, & \text{if } j \notin \tau_{ques}^{'}[: \lfloor \rho/4 \rfloor] \\ 0, & \text{if } j \in \tau_{ques}[: \lfloor \rho/4 \rfloor] \end{cases}, \quad \text{mask}_v[j] = \begin{cases} 1, & \text{if } j \notin \tau_{inter}^{'}[: \lfloor \rho/4 \rfloor] \\ 0, & \text{if } j \in \tau_{inter}^{'}[: \lfloor \rho/4 \rfloor] \end{cases}. \tag{12}$$

$$\tilde{q}_i = \text{mask}_q \cdot q_d', \quad \tilde{v}_i = \text{mask}_v \cdot v_d'. \tag{13}$$

where $\text{mask}_q$ and $\text{mask}_v$ respectively indicate the portions of the question sequence and interaction sequence that require explicit denoising. We also select $\lfloor \rho/4 \rfloor$ data in the sequence for the fusion operation of explicit and implicit denoising to ensure the reliability and diversity of the data.

Additionally, in order to better explore the implicit denoising, we follow the two Transformer encoders used in the baseline: a question encoder $g^Q$ and an interaction encoder $g^V$. These extract embedded representations from given sequences of questions $h_t^Q = g_t^Q(\tilde{q}_{1:t}; m)$ and interactions $h_t^V = g_t^V(\tilde{v}_{1:t}; m)$. The $m$ represents the attention mask controlling the attention modules.

$$h_{t+1}^Q = g_{t+1}^Q(\tilde{q}_{1:t+1}; m_c), \quad h_t^V = g_t^V(\tilde{v}_{1:t}; m_c). \tag{14}$$

where $m_c$ denotes a causal mask having the effect of zeroing out the attention weights of the subsequent positions. Additionally, we employ an extra Transformer encoder, $f^{KR}$ (referred to as the knowledge retriever), to combine the representations of questions and interactions for predicting the next response. Specifically, the knowledge retriever captures relevant questions from the history and references their response results to identify the next response.

$$\tilde{h}_{t+1} = f^{KR}(q = h_{t+1}^Q, k = h_{1:t}^Q, v = h_{1:t}^V; m_c). \tag{15}$$

Where $\tilde{h}_{t+1}$ is the output vector, we concatenate $\tilde{h}_{t+1}$ with $q_{t+1}$ and pass it through a two-layer fully connected network, using the sigmoid function to generate the predicted probability $\hat{r}_{t+1} \in [0, 1]$.

$$\mathcal{L}_{pre} = \sum_t -(r_t \log \hat{r}_t + (1 - r_t) log(1 - \hat{r}_t)). \tag{16}$$

The overall loss for the model is then obtained by combining this SVD-based loss with the primary loss function, resulting in a comprehensive measure of model performance.

$$\mathcal{L}_{total} = \mathcal{L}_{pre} + \eta \cdot \mathcal{L}_{des}. \tag{17}$$

Here $\eta$ is a hyperparameter that we set to 0.01. This will be discussed in the ablation experiments.

## 4 EXPERIMENTS

**Datasets and Baselines** We use four widely-used public datasets to evaluate the performance of the model including **Algebra05**[1], **Algebra06**[1], **Assistment09**[2] and **Slepemapy**[3]. These methods not only include **DKT** (Piech et al., 2015), **DKT+** (Yeung & Yeung, 2018), **DKVMN** (Zhang et al., 2017) which leveraging the deep learning for KT, but also contain **SAKT** (Vaswani, 2017; Pandey & Karypis, 2019),**AKT** (Ghosh et al., 2020), **CL4KT** (Lee et al., 2022) and **DTransformer** (Yin et al., 2023) which leveraged the attention mechanism into KT task.

**Experimental Setup and Results** We adopt the data augmentation parameters from CL4KT for fairness in the experiments. To rigorously evaluate the model's performance, we apply five-fold cross-validation by dividing the data into five subsets and sequentially assessing the model's performance on each subset. We also adopt the baseline strategy of applying early stopping when the AUC on the validation set does not increase over 10 epochs, providing a reliable quantitative evaluation. This experiment is conducted on an NVIDIA 3090 GPU with 24GB of memory.

Table 1 summarizes the evaluation results. After integrating the denoising module, our method achieved the best performance across all four datasets. We also tested other denoising methods on the baseline models: -ID represents implicit denoising, -ED represents explicit denoising, and -DA represents our combined method. The results show that neither implicit nor explicit denoising alone matches the performance of our combined method. Explicit denoising tends to degrade performance due to excessive filtering, while implicit denoising struggles to handle sparse interactions, negatively impacting the representation of knowledge states. Specifically, due to the sparse student interaction data, using only explicit denoising for DKT-ED may lead to over-denoising, especially in a base model like DKT, where the risk of performance degradation is greater. Our method effectively solves these denoising issues, boosting performance.

---

[1]https://pslcdatashop.web.cmu.edu/KDDCup

[2]https://sites.google.com/site/assistmentsdata/home/2009-2010-assistment-data

[3]https://opendatacommons.org/licenses/odbl/1-0/

Table 1: Comparison of AUC and RMSE performance across four datasets for different models.

| Datasets | Algebra05 | | Algebra06 | | Assistment09 | | Slepemapy | |
|---|---|---|---|---|---|---|---|---|
| Metrics | AUC | RMSE | AUC | RMSE | AUC | RMSE | AUC | RMSE |
| DKT | 0.7636 | 0.3921 | 0.7316 | 0.3908 | 0.6891 | 0.4609 | 0.6986 | 0.3978 |
| DKT-ED | 0.7422 | 0.3967 | 0.7165 | 0.3944 | 0.6660 | 0.4675 | 0.6659 | 0.4013 |
| DKT-ID | 0.7642 | 0.3908 | 0.7324 | **0.3908** | 0.6909 | 0.4617 | 0.6992 | 0.4036 |
| DKT-DA | **0.7665** | **0.3896** | **0.7341** | 0.3914 | **0.6917** | **0.4601** | **0.7024** | **0.3961** |
| AKT | 0.7725 | 0.3898 | 0.7474 | 0.3896 | 0.7504 | 0.4438 | 0.7070 | 0.3939 |
| AKT-ED | 0.7936 | 0.3837 | 0.7564 | 0.3850 | 0.7578 | 0.4421 | 0.7630 | 0.3794 |
| AKT-ID | 0.7923 | 0.3831 | 0.7595 | **0.3841** | 0.7567 | 0.4395 | 0.7469 | 0.3849 |
| AKT-DA | **0.7952** | **0.3809** | **0.7633** | 0.3856 | **0.7588** | **0.4388** | **0.7636** | **0.3787** |
| CL4KT | 0.7891 | 0.3815 | 0.7733 | 0.3791 | 0.7624 | 0.4333 | 0.7218 | 0.3926 |
| CL4KT-ED | 0.7969 | 0.3782 | 0.7812 | 0.3749 | 0.7719 | 0.4281 | 0.7487 | 0.3856 |
| CL4KT-ID | 0.7982 | 0.3775 | 0.7921 | 0.3725 | 0.7824 | 0.4241 | 0.7390 | 0.3863 |
| CL4KT-DA | **0.7998** | **0.3766** | **0.7930** | **0.3718** | **0.7834** | **0.4229** | **0.7608** | **0.3795** |

Table 2: To assess robustness, we added Gaussian noise to explicit, implicit, and our method.

| Datasets | Algebra05 | | Algebra05 | | Algebra05 | | Algebra06 | | Algebra06 | | Algebra06 | |
|---|---|---|---|---|---|---|---|---|---|---|---|---|
| Metrics | AUC | RMSE | AUC | RMSE | AUC | RMSE | AUC | RMSE | AUC | RMSE | AUC | RMSE |
| Noise ratio | 0% | | 10% | | 20% | | 0% | | 10% | | 20% | |
| CL4KT | 0.7891 | 0.3815 | 0.7850 | 0.3834 | 0.7809 | 0.3853 | 0.7733 | 0.3791 | 0.7643 | 0.3839 | 0.7568 | 0.3883 |
| CL4KT-ED | 0.7969 | 0.3782 | 0.7573 | 0.3897 | 0.7568 | 0.3902 | 0.7812 | 0.3749 | 0.7456 | 0.3857 | 0.7440 | 0.3873 |
| CL4KT-ID | 0.7982 | 0.3775 | 0.7897 | 0.3800 | 0.7899 | 0.3796 | 0.7921 | 0.3725 | 0.7814 | 0.3784 | 0.7809 | 0.3774 |
| CL4KT-DA | **0.7998** | **0.3766** | **0.7913** | **0.3789** | **0.7904** | **0.3781** | **0.7930** | **0.3718** | **0.7827** | **0.3764** | **0.7819** | **0.3770** |

**Denoising Robustness Analysis** Table 2 illustrates the impact of different denoising methods on model performance under Gaussian noise. Gaussian noise is introduced to simulate the random behavior of students during answering, allowing us to assess the robustness of the model. The experimental results indicate that as the noise intensity increases, the model's performance gradually declines. We compared two datasets with shorter runtimes. For the baseline model CL4KT, performance degrades more significantly as the noise ratio rises, attributed to the model's sensitivity to noisy data, which impacts predictive accuracy. In contrast, our denoising module shows greater stability under varying noise conditions. This is mainly because explicit denoising alone can lead to the loss of important behavioral information crucial for accurate predictions. Implicit denoising reduces the impact of noisy data and better reflects students' true behavioral patterns. However, relying solely on implicit denoising may inadequately address certain low-probability behaviors that significantly affect the student's knowledge state updates. Our combined approach, integrating both explicit and implicit denoising methods, balances these challenges, enhancing model interpretability and stability while ensuring accurate performance. This demonstrates the robustness of our denoising strategy in handling diverse noise levels while maintaining reliable results.

Table 3: This table compares the performance of our denoising method with combined data augmentation versus separate denoising.

| Datasets | Algebra05 | | Algebra06 | | Assistment09 | | Slepemapy | |
|---|---|---|---|---|---|---|---|---|
| Metrics | AUC | RMSE | AUC | RMSE | AUC | RMSE | AUC | RMSE |
| CL4KT-SDS | 0.7929 | 0.3791 | 0.7891 | 0.3789 | 0.7761 | 0.4301 | 0.7548 | 0.3857 |
| CL4KT-FDS | **0.7998** | **0.3766** | **0.7930** | **0.3718** | **0.7834** | **0.4229** | **0.7608** | **0.3795** |

**Data Fusion Comparative Analysis** In this section, we compare different data fusion methods. To validate our method's effectiveness. As shown in the table 3, CL4KT-FDS represents denoising after feature fusion, while CL4KT-SDS indicates denoising the augmented and original sequences separately before feeding them into the model. Combining the augmented sequence with the original sequence and then denoising produces better results compared to denoising the original and augmented sequences separately and then inputting them into the model. This improvement occurs because the augmented and original sequences contain distinct feature information. The fused se-

quence retains the key information from the original sequence while utilizing the additional context and features from the augmented sequence, enabling the model to more comprehensively understand the student's knowledge state. On the other hand, feeding the sequences into the model separately may cause the model to overly rely on data from a single source. Since the original sequence inherently contains noise, the augmented sequence might amplify this noise, making it harder for the model to fully identify and mitigate all noise.

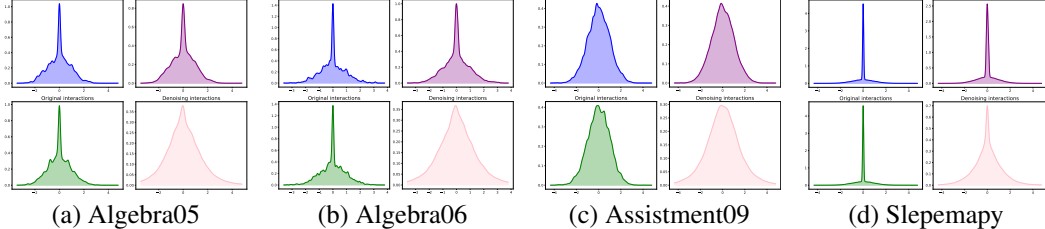

| (a) Algebra05 | (b) Algebra06 | (c) Assistment09 | (d) Slepemapy |

Figure 3: Comparison of feature distributions before and after denoising of questions and interactions in the four datasets.

**Denoising Visualization Analysis** Figure 3 presents the feature distributions of question and interaction sequences across the four datasets. We utilize kernel density plots to visualize these distributions both before and after applying our denoising method. The pre-denoising plots reveal irregular and jagged curves, particularly in the Slepemapy dataset, which may be attributed to lower data similarity and highlights the presence of significant noise or an uneven sample distribution. This irregularity can obscure meaningful patterns and affect the overall analysis. Conversely, the post-denoising plots exhibit much smoother curves with reduced noise interference, suggesting that the data distribution becomes more continuous and closely aligned with the true underlying distribution. This improvement implies that our denoising method effectively reduces noise and better captures the intrinsic patterns, thereby enhancing the quality of the feature representation and the reliability of the subsequent analysis.

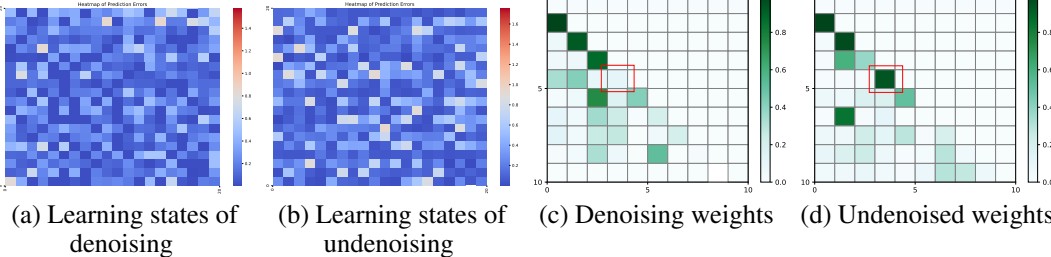

| (a) Learning states of denoising | (b) Learning states of undenoising | (c) Denoising weights | (d) Undenoised weights |

Figure 4: Knowledge state prediction heatmaps and attention visualization, used to compare the impact of historical interactions on future questions with and without denoising.

**Case Studies and Ablation analysis** Figure 4 shows the knowledge state prediction results before and after denoising, as well as the attention weights assigned to the questions and interactions. Comparing (a) and (b), it is observed that the heatmap before denoising has more lighter areas, indicating a larger discrepancy between predicted and actual values, which suggests the presence of noise in the sequence. Our denoising

Table 4: The impact of denoising loss $\mathcal{L}_{des}$ on the model AUC and RMSE in four datasets.

| Datasets | Algebra05 | | Algebra06 | | Assistment09 | | Slepemapy | |
|---|---|---|---|---|---|---|---|---|
| Metrics | AUC | RMSE | AUC | RMSE | AUC | RMSE | AUC | RMSE |
| $\eta = 0$ | 0.7929 | 0.3791 | 0.7891 | 0.3789 | 0.7761 | 0.4301 | 0.7548 | 0.3857 |
| $\eta = 0.01$ | **0.7998** | **0.3766** | **0.7934** | **0.3722** | **0.7833** | **0.4239** | **0.7611** | **0.3793** |

method effectively reduces the impact of this noise. (c) and (d) display the changes in attention weights before and after denoising. For example, in the red-boxed region, a feature has a higher weight before denoising, but becomes lighter after denoising, indicating that the model recognizes it as noise. This shows that after denoising, the model is able to capture underlying patterns in the data and make more accurate predictions.

To validate each component's effectiveness, we compare the impact of denoising loss on the model. The goal is to use it to constrain noise in features, making the data smoother. As shown in the table 4, removing $\mathcal{L}_{des}$ has a noticeable effect on the model, which suggests that constraining noise allows for more accurate identification of anomalies in the sequence.

**Parameter Sensitivity Analysis** To evaluate the denoising effect of augmented sequences, We test $\lambda$ values of 0, 0.01, 0.1 and 0.5. The results indicate that combining augmented data with the original sequence improves performance, but an inappropriate ratio can negatively impact the model. The model primarily relies on real data, with augmented data serving as a supplement. Over-reliance on augmented data may amplify noise and affect the learning process. Therefore, we choose $\lambda$ to be 0.01 for our model.

Table 5: Impact of different $\lambda$ values on model AUC and RMSE across four datasets.

| Datasets | Metrics | 0 | 0.01 | 0.1 | 0.5 |
|---|---|---|---|---|---|
| Algebra05 | AUC | 0.7967 | **0.7998** | 0.7962 | 0.7954 |
| | RMSE | 0.3790 | **0.3766** | 0.3780 | 0.3806 |
| Algebra06 | AUC | 0.7836 | **0.7930** | 0.7821 | 0.7755 |
| | RMSE | 0.3773 | **0.3718** | 0.3766 | 0.3793 |
| Assistment09 | AUC | 0.7815 | **0.7834** | 0.7743 | 0.7539 |
| | RMSE | 0.4260 | **0.4229** | 0.4294 | 0.4346 |
| Slepemapy | AUC | 0.7498 | **0.7608** | 0.7433 | 0.7417 |
| | RMSE | 0.3862 | **0.3795** | 0.3886 | 0.3898 |

Table 6 presents the parameter selection for the division of explicit denoising samples. We test $\rho$-value coefficients of 0, 0.25, 0.5 and 1. A coefficient of 0 indicates no explicit denoising, while a coefficient of 1 indicates no implicit denoising, effectively creating a spectrum of denoising approaches. After conducting multiple experiments, we determine that a $\rho$-value coefficient of $1/4$ yielded the best performance, as it struck an optimal balance between explicit and implicit denoising. This selection not only significantly improves model accuracy but also enhances the overall interpretability and robustness of the results.

Table 6: Comparison of the effects of different $\rho$ coefficients in four datasets.

| Datasets | Algebra05 | | Algebra06 | | Assistment09 | | Slepemapy | |
|---|---|---|---|---|---|---|---|---|
| Metrics | AUC | RMSE | AUC | RMSE | AUC | RMSE | AUC | RMSE |
| 0 | 0.7982 | 0.3775 | 0.7921 | 0.3725 | 0.7824 | 0.4241 | 0.7390 | 0.3863 |
| $\rho/4$ | **0.7998** | **0.3766** | **0.7930** | **0.3718** | **0.7834** | **0.4229** | **0.7608** | **0.3795** |
| $\rho/2$ | 0.7947 | 0.3793 | 0.7839 | 0.3780 | 0.7761 | 0.4301 | 0.7472 | 0.3859 |
| $\rho$ | 0.7969 | 0.3782 | 0.7812 | 0.3749 | 0.7719 | 0.4281 | 0.7487 | 0.3856 |

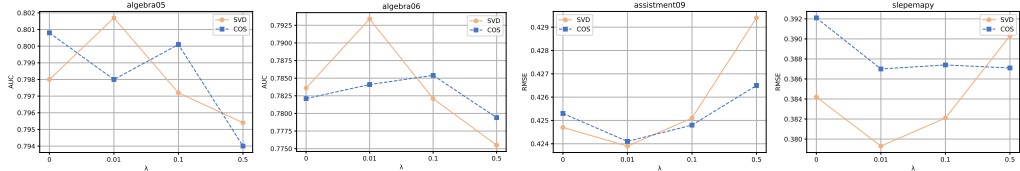

Figure 5: Comparison of AUC and RMSE of different denoising sample selection methods on different datasets.

**Denoising Sample Selection Strategy** As shown in Figure 5, we compared SVD-based and similarity-based denoising methods for fusing augmented and original sequences. SVD outperforms the similarity-based method, which requires manual sample adjustments, leading to instability and lower interpretability. SVD, on the other hand, adapts sample size automatically, improving performance and interpretability.

## 5 CONCLUSIONS

In this paper, we present a plug and play framework for KT that incorporates a denoising module. To address the noise problem and interaction sparsity, we apply both explicit and implicit denoising during the data augmentation preocess, effectively reducing noise in both augmented and original data, which enhances sequence representations. Our comprehensive experiments demonstrate that our model significantly outperforms current state-of-the-art methods in terms of prediction accuracy and data representation quality.

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
