# OpenReview forum: "Sequence Denoising with Self-Augmentation for Knowledge Tracing"
_ICLR.cc/2025/Conference — Submitted to ICLR 2025_

### Official Review · Reviewer_hBSQ · 2024-10-22

**Soundness:** 2
**Presentation:** 2
**Contribution:** 2
**Rating:** 3
**Confidence:** 4

**Summary:**

This study addresses the issue of noise in both original and augmented sequences within the field of knowledge tracing and proposes a self-enhanced sequence denoising knowledge tracing algorithm (SDAKT). The effectiveness of the algorithm is validated through comparison with three baseline algorithms. Further experimental results show that, compared to models without any denoising operations, SDAKT is more robust to noise.

**Strengths:**

1. The motivation of addressing noises in knowledge tracing is well explained.
2. The experiments have shown certain improvements especially regarding addressing noises.

**Weaknesses:**

1. The algorithm design has not been adequately explained. The rationale for using Equation (8) to quantify the noise in real data is not thoroughly explained. Since the singular vectors of the matrices before and after denoising differ, directly comparing the singular values seems questionable and may not provide a meaningful measure of the noise.

2. The algorithm's performance is not convincing. Although the denoising-enhanced version of the algorithm shows some improvement in AUC and RMSE metrics compared to baselines, the gains are relatively modest. Additionally, the algorithm includes both explicit and implicit denoising mechanisms. However, as seen in Table 1, the performance of the SDAKT algorithm does not significantly differ from using only one of these denoising strategies. This suggests that the combination of both denoising approaches does not substantially enhance the overall performance. One of the key innovations of this paper is the fusion of explicit and implicit denoising strategies, yet the results indicate that this fusion does not demonstrate clear advantages in practice.

3. Many writing and presentation problems: The paper suffers from inconsistencies in formatting and imprecise language. For example, in Equations (6) and (7), embedding vectors are inappropriately subjected to singular value decomposition, when matrix forms should have been used. Additionally, the citation format in line 196 is incorrect, and the reference formatting is inconsistent throughout the paper. Finally, there is a discrepancy between the description and Equation (4) in the text: while the description refers to the original sequences, the equations present denoised sequences instead.

**Questions:**

I would appreciate the authors to respond to weaknesses mentioned above.

---

> ### Author Response · Authors · 2024-11-25
> **Response to Reviewer hBSQ**
>
> We thank the reviewer for their feedback and address their questions below.
>
> W1:The algorithm design has not been fully explained. In equation (8), the principle of quantifying noise in real data sets has not been thoroughly explained. Direct comparison of singular values does not seem reasonable enough and may not provide a meaningful noise measurement.
>
> Thank the reviewers for their valuable comments. In the formula, we used the binary norm of the singular value matrix to represent the difference between the problem and the interaction sequence before and after denoising, and the larger the difference value, the lower the similarity between the features, indicating that the part of data may contain noise. In this way, we detected noise and classified the data. Regarding the direct comparison of singular values mentioned by the reviewers, we acknowledge that this method may raise questions, but we emphasize that the calculation of this difference is to effectively evaluate the impact of noisy data, not to directly make a simple comparison of singular values, and we will further clarify this point to make the description more rigorous. Thanks again to the reviewers for their careful review and valuable comments.
>
> W2:The algorithm's performance is unconvincing, with modest improvement gains. While it integrates explicit and implicit denoising mechanisms, Table 1 shows no significant advantage over using either strategy alone. This fusion, a key innovation, fails to demonstrate clear practical benefits.
>
> Thank you for the reviewer’s comments. Our explanation is as follows:
>
> While Table 1 shows relatively limited performance gains, we would like to emphasize that the main advantage of this combination lies not only in the improvement in performance but also in the enhancement of interpretability. In the paper, we have outlined the limitations of using explicit or implicit denoising alone. Explicit denoising helps identify and handle obvious noise points in the data, while implicit denoising better captures differences in students’ response patterns, thereby improving the overall robustness of the data.
> Moreover, we conducted robustness experiments on using a single denoising strategy, as shown in Table 2, where the results indicate greater instability with a single approach. We will further elaborate on the motivation behind this strategy in the paper.
>
> W3:The paper has a number of writing and presentation problems, including formatting inconsistencies and language inaccuracies. For example, in equations (6) and (7), the embedding vector was improperly affected by singular value decomposition when matrix form should have been used, the citation format in line 196 was incorrect, and the reference format was inconsistent. In addition, there is a mismatch between the description and equation (4), where the text refers to the original sequence, while the formula uses the denoised sequence.
>
> Thank you for the reviewer’s suggestions. We will thoroughly review the paper’s formatting and language to ensure consistency and accuracy. Regarding the equations, we acknowledge that our expression and descriptions may lack precision, and we will make the necessary improvements. For the discrepancy between the text description and equation (4), we combined the original and denoised sequences to create a new sequence, which might not have been clearly explained in the text. We will clarify this in the revised version. Once again, we sincerely thank you for your detailed review and valuable feedback, which will greatly help us improve the quality and readability of our paper.

---

### Official Review · Reviewer_4SAi · 2024-11-02

**Soundness:** 3
**Presentation:** 1
**Contribution:** 2
**Rating:** 5
**Confidence:** 3

**Summary:**

This paper studies knowledge tracing from the perspective of denoising original interaction sequences. To measure the degree of outlier, singular values from Singular Value Decomposition (SVD) are used.

**Strengths:**

(1) Singular values are used to measure the degree of outliers in interaction sequences.

(2) Soft and hard denoising strategies are proposed. Soft corresponds to maximize the maximum singular value. Hard corresponds to explicitly mask some nosiy examples.

(3) The experiments are conducted on three backbones to test the effectiveness of the proposed method.

**Weaknesses:**

(1) Explicit denosing seems to not be very effective. As shown in Table 1, DKT-ED performs much worse than DKT. Moreover, the combination of explicit denoisng and implicit denoising is not very beneficial, which is also reflected in Table 1.

(2) Some parts of this paper are questionable. For example, why is SVD performed for vectors, as shown in Eq 6 and 7? “Question q1 was answered incorrectly three times at first and correctly two times latter” is not consistent with the content in Figure 1. Why f_{den} can play a role of removing noise is not introduced. Why using data augmentation in the denoising part?

(3) Some important details are missing. What is the meaning of q_d? Figure 2 is not clear. How to calculate the influence weights in “in traditional KT methods, the influence weights” is not explained and the references are not mentioned. The effect of lambda is not very reasonable, as shown in Figure 5.

**Questions:**

Why maximizing the largest singular value can reduce noise.

---

> ### Author Response · Authors · 2024-11-25
> **Response to Reviewer 4SAi**
>
> We thank the reviewers for their valuable feedback. We address these issues below and add to the manuscript by adding more clarifications and new research.
>
> W1:As Table 1 shows, DKT-ED performs much worse than DAT, and the combination of explicit and implicit denoising is not as good.
>
> In the field of knowledge tracking (KT), due to the relatively sparse student interaction data, explicit denoising alone may cause excessive denoising problems and affect model performance. In this case, explicit de-noising can mistakenly delete normal interactive data, especially in base models such as DKT that do not use additional information, and the risk of performance degradation is more significant. We have detailed the reasons for this performance degradation in our analysis in Table 1. In addition, for the understanding of excessive Denoising, we refer to the related research SSDRec: Self-Augmented Sequence Denoising for Sequential Recommendation. Specifically, explicit denoising can effectively reduce noise interference to the data, while implicit denoising helps the model automatically adapt to different noise patterns during the inference process. Through this combination, our model not only improves the accuracy, but also enhances the interpretability of the model.
>
> W2:Why SVD vectors, as shown in formulas 6 and 7? The first three wrong answers to q1 and the last two correct answers are inconsistent with the contents in Figure 1. Why can f_{den} play the role of denoising? This paper does not introduce why data enhancement is needed for denoising?
>
> 1.(6) and (7) use SVD to decompose problem vectors and interaction vectors to reduce noise effects in data representation. By retaining the main singular values, we can effectively reduce the noise component, and thus more clearly identify and distinguish the noisy data that may be present in the original sequence.
>
> 2.Thanks to the reviewer for pointing out that Figure 1 is inconsistent with the description of "q1 answer wrong for the first three times and correct for the last two times". We will make corrections to Figure 1 to ensure that the example is consistent with the diagram and to avoid confusion for the reader.
>
> 3.For the denoising mechanism of f_{den}, we refer to SSDRec: Self-Augmented Sequence Denoising for Sequential Recommendation and Hierarchical Item Inconsistency Signal Learning for sequential recommendation Sequence Denoising in Sequential Recommendation, and some denoising modules are used for optimization. The revised draft will add a detailed introduction to the mechanism and principle of f_{den) denoising.
>
> 4.In addition, given the prevalence of data sparsity in the field of knowledge tracking (KT), there are existing methods to mitigate this problem through data enhancement. But in our study, there was noise in the original data set, and doing data enhancement directly could amplify that noise. Therefore, while improving the diversity of data, we de-noised the original data and enhanced data to further improve the data quality on the basis of rich data.
>
>
> W3:Some important details are missing. What is the meaning of q_d? Figure 2 is not clear. How to calculate the influence weights in “in traditional KT methods, the influence weights” is not explained and the references are not mentioned. The effect of lambda is not very reasonable, as shown in Figure 5.
>
> Thank the reviewers for their detailed review of our work and constructive comments. In response to these questions, we provide the following responses, which will be supplemented in the revised draft.
>
> 1.q_{d} represents the new feature representation vector of the problem sequence after denoising, and v_{d} represents the new feature representation vector of the interaction sequence after denoising. We will explain these symbols in detail in the revised draft to ensure that the symbols are more clearly defined.
>
> 2.Regarding the clarity of Figure 2, we will describe Figure 2 in more detail, adding comments and annotations to convey the information more intuitively.
>
> 3.As for the calculation of influence weights, we have demonstrated them in visualizations and referred to relevant literature. Tracing Knowledge Instead of Patterns: Stable Knowledge Tracing with Diagnostic Transformer to illustrate the effectiveness of our method for weight analysis. Through the analysis of experimental results, we further explain why our method is more efficient in weight calculation.
> 4.Regarding the design of the λ parameter, we have designed this parameter to adjust the ratio of the original sequence to the enhanced sequence. Since the model is more reliable to the original sequence, we explore four different λ values in the experiment. Based on the results in Figure 5, we find that the best performance is achieved on average across the four data sets when λ is 0.01. These values obtained in the experiment will help to better understand the role of λ.

---

> > ### Comment · Reviewer_4SAi · 2024-12-03
> >
> > Thanks for the response and I will keep my scores.

---

> ### Author Response · Authors · 2024-11-25
> **Response to Reviewer 4SAi**
>
> Thanks to the reviewers for their in-depth attention to our method, we explain the question about maximizing the maximum singular value to reduce noise as follows:
>
> The core idea of maximizing the maximum singular value is to remove low-information noise components by preserving the most important information in the data. In singular value decomposition (SVD), the data matrix is decomposed into the product of three matrices: U, Σ, and V^⊤. The size of the singular value Σ reflects the importance or information of the data, where larger singular values correspond to the most representative part of the data, while smaller singular values usually represent noise and unimportant components. By maximizing the maximum singular value, we are actually selecting the primary components that best represent the data, and by compressing or ignoring smaller singular values, removing those low-frequency components that might add to the noise. This processing method can effectively reduce the influence of noise, and improve the quality of data and the robustness of the model. We will add a detailed explanation of this theoretical background in the revised version to help readers better understand the role of maximizing the maximum singular value. Thank you again for your valuable questions.

---

### Official Review · Reviewer_3JH6 · 2024-11-04

**Soundness:** 3
**Presentation:** 3
**Contribution:** 2
**Rating:** 3
**Confidence:** 4

**Summary:**

This paper introduces SDAKT that addresses noise in both original and augmented sequences using explicit and implicit denoising techniques. The main innovation is combining explicit and implicit denoising approaches during the data augmentation process, using SVD to balance between hard and soft noise reduction. The method denoises both original and augmented sequences to improve feature representation.

**Strengths:**

- This paper studies the important problem of noisy interactions and sparse distribution in knowledge tracing data.
- The paper structure of the solution is clear and easy to understand.

**Weaknesses:**

- The motivation for noise in KT interaction sequences is poorly defined. The paper doesn't clearly quantify the extent of noise in real KT datasets or demonstrate its impact empirically.
- This article was unable to find comparison results for other baselines.
- The artificial noise injection experiments only use Gaussian noise, which may not reflect realistic noise patterns in educational data.
- There may be confusion between the model names SDAKT and CL4KT-DA.
- Limited analysis of computational overhead introduced by the denoising module.

**Questions:**

1 What is “unreliable knowledge states” (line 55) ？

2 Why KT datasets sparsity problems amplify noise?

3 “three errors, two correct” in (Line 65 q1) is inconsistent with Figure 1.

4 Why is the performance of the sequence model DKT-ED worse than the original DKT?

5 Why is variable Gaussian noise used in Table 2? How does variable noise affect the KT model?

---

> ### Author Response · Authors · 2024-11-25
> **Response to Reviewer 3JH6**
>
> We are grateful for the strong evaluation and detailed feedback on our analyses.
>
> W1:The motivation for noise in KT interaction sequences is poorly defined. The paper doesn't clearly quantify the extent of noise in real KT datasets or demonstrate its impact empirically.
>
> Thank the reviewers for their valuable comments. We reply as follows:
>
> In the KT field, noise often comes from student guesswork, slips, or occasional wrong reactions, factors that are often present in the actual data but difficult to quantify accurately. Therefore, we did not directly quantify the degree of noise in the experiment, but only manually injected different degrees of noise and observed its impact on the accuracy of the model prediction.
>
> W2:This article was unable to find comparison results for other baselines.
>
> Thank the reviewers for their valuable comments. We reply as follows:
>
> We understand the reviewers' concern with baseline comparison results. Since our proposed approach is a plug and play module, in our study we aim to be compatible with the existing KT model, focusing on the effectiveness of the module, without presenting more baseline model comparisons in the paper.
>
> W3:The artificial noise injection experiments only use Gaussian noise, which may not reflect realistic noise patterns in educational data.
>
> Thanks to the reviewer for his comments on the artificial noise injection experiment. We explain as follows:
>
> In the experiment, we chose to use Gaussian noise for noise injection in order to evaluate the robustness of the model under common noise conditions. We understand the reviewer's point of view that the real noise pattern in education data may be more complex, not limited to Gaussian noise. In order to further verify the model's performance in more realistic scenarios, we plan to introduce other types of noise such as random guess or systematic bias in future studies, so as to reflect the noise characteristics in education data more comprehensively.
>
> W4:There may be confusion between the model names SDAKT and CL4KT-DA.
>
> Thanks to the reviewer for pointing out possible confusion between SDAKT and CL4KT-DA. In order to avoid confusion among readers, we will explicitly and uniformly use CL4KT-DA as the abbreviation of the model in the paper, and make clarifications and adjustments in relevant parts to ensure consistency and clarity of terms.
>
> W5:Limited analysis of computational overhead introduced by the denoising module.
>
> Thanks for the reviewer's concern about the computational cost of denoising model, we reply as follows:
>
> In our study, due to the small dimension of the input data of the model, the introduction of the denoising module does not significantly increase the computing overhead. At this data scale, the impact of the denoising module on the overall training and reasoning time is negligible.

---

> > ### Author Response · Authors · 2024-11-25
> > **Response to Reviewer 3JH6**
> >
> > We are grateful for the strong evaluation and detailed feedback on our analyses.
> >
> > Q1:What is “unreliable knowledge states” (line 55) ?
> >
> > Thank the reviewer for raising the "question about the unreliable state of knowledge". We explain as follows:
> >
> > An unreliable state of knowledge means that the level of student knowledge inferred by the model may be unstable or inaccurate due to noise or other factors in the data. For example, students' accidental speculation or carelessness in answering questions will lead to the model's misjudgment of their actual knowledge state, which cannot fully reflect the state of students' real knowledge mastery and affect the final prediction result.
> >
> > Q2: Why KT datasets sparsity problems amplify noise?
> >
> > Thank you for your questions about sparsity and noise amplification. We explain as follows:
> >
> > In the field of KT, student interaction data is usually sparse, as each student has a limited record of answers. This sparsity makes the model have to rely on limited interactive information to infer the knowledge state of students and make predictions. However, these limited interactive data may contain noise data such as guesses and careless errors, which can cause the model to misjudge the actual knowledge level of students.
> > When data enhancement is performed on the original sequence, the existing noisy data may be copied and amplified, thus increasing its proportion in the overall data set. This makes the model more susceptible to noisy data during training, which amplifies the adverse effect of noise on the model's predictive performance. Therefore, the sparsity problem amplifies the effect of noise because in the case of sparse data, the proportion of noise has a greater impact on the stability and accuracy of model predictions.
> >
> > Q3:three errors, two correct” in (Line 65 q1) is inconsistent with Figure 1.
> >
> > Thanks to the reviewer for pointing out that Figure 1 is inconsistent with the description of "Question q_1 answered wrong three times and correct the last two times". We will make changes to Figure 1 to ensure that the example is consistent with the diagram and to avoid confusion for the reader. Thanks again for the reviewer's valuable suggestions.
> >
> > Q4:Why is the performance of the sequence model DKT-ED worse than the original DKT?
> >
> > Thank the reviewers for their valuable feedback. We explain as follows:
> >
> > Due to the relatively sparse student interaction data, the fact that DKT-ED uses only explicit de-noising can lead to excessive de-noising problems, especially in basic models like DKT that do not use additional information, and the risk of performance degradation is more significant. We explain the reasons for the performance decline in our analysis in Table 1.
> >
> > Q5:Why is variable Gaussian noise used in Table 2? How does variable noise affect the KT model?
> >
> > Thanks for the reviewer's question about the use of variable Gaussian noise in Table 2. We explain as follows:
> >
> > In the experiment, we used Gaussian noise to simulate the data fluctuations introduced by random behavior (guessing or slipping) in students' answers. By adding Gaussian noise to the data, we could test the performance of the model in the face of different levels of noise and verify the robustness of the model. The noise of different variables is used for comparison. Generally speaking, the greater the proportion of noise, the worse the reliability of the data, and the effect of the model will decline. Our experimental results show that the performance of the model in a high-noise environment has a smaller decline, showing a stronger robustness.

---

> > ### Comment · Reviewer_3JH6 · 2024-11-27
> > **Response to rebuttal**
> >
> > Some of my questions were answered, but it is inappropriate to say that students' guesses, mistakes, or occasional incorrect responses are difficult to quantify and model (W1 rebuttal). This paper studies these factors as “noise”, but as early as in the BKT model, these factors have been clearly defined and quantified through parameters [1,2]. These parameters can be learned from real data and have been widely proven to be effective.
> >
> > In addition, if there is no baseline comparison, I think it would be more appropriate to delete the description of baselines in lines 283-284.
> >
> > [1] Parametric Constraints for Bayesian Knowledge Tracing from First Principles. EDM 2024.
> >
> > [2] Improving Model Fairness with Time-Augmented Bayesian Knowledge Tracing. LAK 2024.

---

> > > ### Author Response · Authors · 2024-11-27
> > > **Response to Reviewer 3JH6**
> > >
> > > Thank you very much for taking the precious time to reply to us.
> > >
> > > We fully recognize that in the BKT model, factors such as students' guessing and slipping have been clearly defined and quantified through specific parameters as you pointed out. However, in our research, regarding these factors as "noise" doesn't mean denying the existing quantification methods in previous models. Instead, it is based on a slightly different research perspective and objective. We aim to explore the impact of these "noise"-like elements on the overall learning process and results in teaching scenarios that are more complex, diverse and full of interference from many real-life situations. For example, in the data of actual scenarios, there may be situations like students with different learning styles frequently switching their learning methods and sudden short-term distractions caused by the external environment leading to group distraction among students. The uncertainties arising from these situations are difficult to accurately fit simply by applying the parameters of the BKT model. Therefore, we attempt to consider their comprehensive impacts from the perspective of "noise" as a whole.
> > >
> > > We are extremely grateful for the feedback you provided regarding the description in lines 283 to 284. We will immediately re-examine that part and, following your suggestion, delete the relevant description to ensure the clarity and coherence of the manuscript and avoid causing any potential confusion to readers. Once again, thank you for taking the time to review our work. We will surely strive to improve the paper based on your comments.

---

### Official Review · Reviewer_cRkc · 2024-11-04

**Soundness:** 2
**Presentation:** 2
**Contribution:** 2
**Rating:** 5
**Confidence:** 4

**Summary:**

The paper presents a Sequence Denoising with Self-Augmentation for Knowledge Tracing (SDAKT) model aimed at improving knowledge tracing by reducing the impact of noise in students' interaction sequences through explicit and implicit denoising techniques, leveraging Singular Value Decomposition (SVD) for both noise detection and feature extraction. Experimental results show significant improvements in model robustness and predictive accuracy across multiple datasets, indicating the efficacy of the proposed denoising framework.

**Strengths:**

- An approach to handling noise in knowledge tracing through two-stream denoising.
- Utilize SVD for explicit and implicit denoising, improving robustness and accuracy.
- Demonstrate performance gains across different standard datasets.

**Weaknesses:**

- The paper overlooks some important baselines. For instance, HD-KT [1], a relevant method that also addresses denoising for guessing and slipping issues, is not discussed or compared, which limits the contextual understanding of the model's contributions. Additionally, more works in sequence denoising, as mentioned in the Related Work section, should be considered as baselines to better validate the effectiveness of the proposed method.

- The main technical contribution of the paper is the use of SVD decomposition to analyze informative signals. However, this approach is relatively simple and has been applied in other fields. Additionally, the paper lacks a thorough discussion on computational overhead, particularly the time-intensive nature of SVD calculations, which could impact real-time feasibility in large-scale applications.

- There are minor writing issues: Missing punctuation at the end of the formula; inconsistent of reference format and so on.

[1] HD-KT: Advancing Robust Knowledge Tracing via Anomalous Learning Interaction Detection. Proceedings of the ACM on Web Conference 2024.

**Questions:**

- Could this paper compares the proposed method with KT methods that also perform denoising or with approaches from sequence denoising to verify the effectiveness of the proposed approach?

- In the ablation study, I noticed that the performance of CL4KT-DA and CL4KT-ID on Algebra06 is identical. Could this paper explain the reason for this?

---

> ### Author Response · Authors · 2024-11-25
> **Response to Reviewer cRkc**
>
> We are grateful for the strong evaluation and detailed feedback on our analyses.
>
> W1:This article ignores some important benchmarks. For example, HD-KT is not discussed or compared, and, as mentioned in the related work section, more sequence denoising work should be considered as a benchmark to better validate the effectiveness of the proposed methods.
>
> Thank the reviewers for their valuable comments. We will expand the related work section to cover more research and methods in sequence denoising. Thanks again for the reviewer's suggestions, we will improve the manuscript according to the comments.
>
> W2:The main technical contribution of the paper is the use of SVD decomposition to analyze informative signals. However, this approach is relatively simple and has been applied in other fields. Additionally, the paper lacks a thorough discussion on computational overhead, particularly the time-intensive nature of SVD calculations, which could impact real-time feasibility in large-scale applications.
>
> Thank you for your questions. We acknowledge that SVD decomposition is a classical method and has been widely used in other fields. However, the use of SVD for KT domain recognition and noise reduction has been less explored. As for the computational overhead, our research focuses on educational data sets, which are usually relatively small in size, so no significant time overhead problem is seen in this study. We will explore the impact of time overhead on datasets of different sizes in future studies, and make specific analysis for large-scale datasets.
>
> W3:There are minor writing issues: Missing punctuation at the end of the formula; inconsistent of reference format and so on.
>
> Thank you for the questions raised by the reviewer. I am very sorry for this mistake. We will carefully review and correct handwriting and formatting problems in the manuscript to improve the overall quality and readability of the article. Thanks to the reviewers for their careful review and valuable feedback.

---

> > ### Author Response · Authors · 2024-11-25
> > **Response to Reviewer cRkc**
> >
> > We thank the reviewers for their valuable feedback, which enabled us to improve the manuscript and give a more accurate description of the detailed aspects of the paper with additional experiments that included new baselines and more data, as well as notes on methods, interpretability and case studies.
> >
> > Q1:Could this paper compares the proposed method with KT methods that also perform denoising or with approaches from sequence denoising to verify the effectiveness of the proposed approach?
> >
> > Thank the reviewers for their valuable comments. We recognize that comparing the proposed method with other KT methods or sequence denoising methods with de-noising functions can help to verify the effectiveness and contribution of the method more comprehensively, and we will supplement the experimental results in the future for comparison. Thanks again for the reviewer's suggestions, we will improve according to your feedback to enhance the integrity and persuasiveness of the article.
> >
> > Q2:In the ablation study, I noticed that the performance of CL4KT-DA and CL4KT-ID on Algebra06 is identical. Could this paper explain the reason for this?
> >
> > Thanks to the reviewer for raising this question. I'm very sorry for this mistake. In fact, these two methods should show different effects in the experiment. We will carefully review the experimental data and correct the errors here, and will further review other data to avoid similar issues.

---

> > > ### Comment · Reviewer_cRkc · 2024-12-03
> > > **Thanks for the feedback**
> > >
> > > I have read the rebuttal. Thanks.

---

### Meta-Review · Area_Chair_b54K · 2024-12-10

**Metareview:**

The technical contribution could be considered straight forward and the injection of noise is limited to Gaussian noise. The paper would \ benefit from clearer definitions and a better motivation why the proposed approach is general enough and also tailored to the domain (e.g., why are Gaussians the right distribution to draw from?). It also seems that the proposed approach is not always supported by empirical evidence, the paper needs more in-depth empirical evaluations and appropriate baselines are missing.

**Additional Comments On Reviewer Discussion:**

Some reviewers engaged in discussion with the authors and commented on the author responses. However, this is rather a clear case, nobody was really excited about the paper in the first place as reflected by the reviews and, although they did well, responses by authors cannot change the big picture.

---

### Decision · Program_Chairs · 2025-01-22

Reject